# Atomic and Molecular Layer Deposition as Surface Engineering Techniques for Emerging Alkali Metal Rechargeable Batteries

**DOI:** 10.3390/molecules27196170

**Published:** 2022-09-20

**Authors:** Matthew Sullivan, Peng Tang, Xiangbo Meng

**Affiliations:** Department of Mechanical Engineering, The University of Arkansas, Fayetteville, AR 72701, USA

**Keywords:** atomic layer deposition, molecular layer deposition, alkali metals, surface coating

## Abstract

Alkali metals (lithium, sodium, and potassium) are promising as anodes in emerging rechargeable batteries, ascribed to their high capacity or abundance. Two commonly experienced issues, however, have hindered them from commercialization: the dendritic growth of alkali metals during plating and the formation of solid electrolyte interphase due to contact with liquid electrolytes. Many technical strategies have been developed for addressing these two issues in the past decades. Among them, atomic and molecular layer deposition (ALD and MLD) have been drawing more and more efforts, owing to a series of their unique capabilities. ALD and MLD enable a variety of inorganic, organic, and even inorganic-organic hybrid materials, featuring accurate nanoscale controllability, low process temperature, and extremely uniform and conformal coverage. Consequently, ALD and MLD have paved a novel route for tackling the issues of alkali metal anodes. In this review, we have made a thorough survey on surface coatings via ALD and MLD, and comparatively analyzed their effects on improving the safety and stability of alkali metal anodes. We expect that this article will help boost more efforts in exploring advanced surface coatings via ALD and MLD to successfully mitigate the issues of alkali metal anodes.

## 1. Introduction

Since commercialized in 1991 by Sony, lithium-ion batteries (LIBs) have dominated portable electronic devices and have even started to penetrate the market of electric vehicles [1]. With their continuous development, LIBs now are approaching their theoretical energy limits [2,3,4,5,6]. Thus, it is urgent to develop new battery technologies and materials for further boosting energy density, extending lifetime, improving safety, and reducing cost [7,8].

In pursuing next-generation battery technologies with higher energy density, alkali metals are promising as anodes, due to their high capacity or abundance. Specifically, lithium (Li), sodium (Na), and potassium (K) have capacities of 3861, 1166, and 685 mAh/g, respectively. In addition, their potentials are −3.04, −2.73, and −2.936 V versus the standard hydrogen electrode (SHE), respectively. Li metal enables the highest theoretical specific capacity and the lowest electrochemical potential [9,10,11,12,13]. Thus, Li metal has been considered as a “holy grail” for emerging lithium metal batteries (LMBs), such as lithium-sulfur (Li-S) and lithium-oxygen (Li-O_2_) batteries [11,14,15]. In comparison, Na metal is abundant and cost-effective. It can be used to constitute a variety of sodium metal batteries (NMBs), such as Na-S [16], Na-O_2_ [17], and Na-CO_2_ batteries [18,19]. K metal can enable a capacity about two times that of graphite anodes in state-of-the-art LIBs. Despite their great potentials as anodes in alkali metal batteries (AMBs), they are plagued by stability and safety issues, mainly due to two reasons: unstable solid electrolyte interphase (SEI) and dendrite growth [20]. The ionically conductive but electronically insulating SEI layer was first coined by Pealed et al. in 1979 [21,22]. It originates from the spontaneous reactions between alkali metal anodes and organic liquid electrolytes [23]. The SEI layer is generally non-uniform and heterogeneous in composition and property [24]. During a plating process, furthermore, the generated alkali metal dendrites can easily break the fragile and soft SEI, exposing new fresh alkali metal to the electrolyte and thereby leading to more SEI formation [25]. In the subsequent stripping process, the gracile alkali dendrites could be trapped and electrically isolated within the SEI layer. This results in dead alkali metals [26,27,28]. As the plating/stripping processes repeat, electrolyte and active alkali metals are continuously consumed. At the same time, cell impedance increases with inferior Coulombic efficiency (CE) [9,29]. Particularly, the propagation of alkali metal dendrites can penetrate the cell’s separator, lead to electrical short of cells, and cause battery thermal runaway, fires, and even explosions, as illustrated in Figure 1 [10,13,30].

To address these issues of AMBs, many strategies have been explored for Li [31,32], Na [33], and K [34] metals, including solid-state electrolytes [35,36,37,38,39,40], new electrolytes and additives [41,42,43], ionic liquids [44,45,46], and nanostructured electrode designs [47,48]. Some comprehensive reviews of these techniques have been documented in literature [33,49,50,51,52,53]. Recently, atomic and molecular layer deposition (ALD and MLD) have emerged as two tremendously powerful techniques for surface engineering of alkali metal anodes, featuring high accuracy over material growth, low process temperature, and extremely uniform and conformal coverage over substrates of any shape [8,54,55]. Ascribed to their unique growth mechanisms, ALD and MLD could constitute the most intimate integration between the resultant coating and the substrate, thereby minimizing interfacial issues due to imperfect contacts. Thus, they remain as one of the most promising techniques for addressing the issues of alkali metal anodes. The effectiveness of ALD and MLD coatings depends on their resulting properties (e.g., electrical conductivity, ionic conductivity, and mechanical properties). A desirable coating should be electrically insulating, ionically conductive, mechanically robust, and chemically and electrochemically stable [15,56]. In this regard, our recent MLD coating of lithium glycerol (LiGL) has exhibited very promising performance [57]. Thus, ALD and MLD have emerged as two novel tools for developing AMBs. However, to date there lacks a comprehensive survey on the efforts of ALD and MLD for superior alkali metal anodes. To this end, in this review, we have made a thorough summary on ALD and MLD coatings of alkali metal anodes.

Following this introductory section, we will firstly brief the mechanisms and characteristics of ALD and MLD. Then, we will focus on summarizing ALD inorganic coatings, MLD organic coatings, and ALD/MLD dual layered surface modification in three sections. In the last part, we conclude this work and provide some outlook on future research. We expect that this work will provide readers an integral picture on the surface engineering ALD and MLD in development of next-generation AMBs and further stimulate more efforts on advanced surface modifications via ALD and MLD.

## 2. Mechanisms and Characteristics of ALD and MLD

Since its introduction in the 1970s [58,59], ALD has now recognized as a powerful technique for surface and interface engineering in energy-related devices, as is well documented in literature [2,8,54,59,60,61,62,63,64,65,66,67]. One of the most successful and prevalent ALD processes is growing Al_2_O_3_ films with the precursors of trimethylaluminum (TMA) and H_2_O. Two half surface reactions are proceeded as follows [65,68].
|-OH + Al(CH_3_)_3_↑→|-O-Al(CH_3_)_2_ + CH_4_↑(1)
|-O-Al(CH_3_)_2_ + 2H_2_O↑→|-OAl(OH)_2_ + 2CH_4_↑(2)
where “|” indicates the substrate surface, “-” indicates a chemical bond, and “↑” denotes the gas phase of the precursors (i.e., TMA and water) and the byproduct (i.e., CH_4_). Typically, a four-step procedure is required to finish one ALD cycle: (i) pulsing the first precursor of TMA to react with the surface -OH functional groups and thereby produce a new layer of -Al(CH_3_)_2_ bounded to oxygen, releasing CH_4_ as the byproduct, as shown in Equation (1); (ii) purging with an inert gas (e.g., Ar) to remove the oversupplied precursor TMA and the resultant byproduct CH_4_; (iii) pulsing the second precursor H_2_O to restore the substrate surface back to a full coverage of -OH with the release of CH_4_ as the byproduct, as given in Equation (2); and (iv) purging with the inert gas to remove the oversupplied H_2_O and the byproduct CH_4_. A visual representation of this reaction and its analogous inorganic-organic hybrid aluminum ethylene glycol (AlEG) is presented in Figure 2. These two half surface reactions (i.e., Equations (1) and (2)) are self-limiting or self-terminating due to the finite amount of surface functional groups (e.g., -OH and -CH_3_). The Al_2_O_3_ film thickness usually can be precisely controlled at the atomic level [65,69], e.g., a growth per cycle (GPC) of ~1 Å [70]. More details about the chemistry of ALD can be found in some previous excellent reviews [71,72,73,74]. ALD is exclusively for growing inorganic thin films ranging from binary [74,75,76,77], ternary [78], and even more complex compounds [79,80,81,82] MLD shares a highly similar working principle as that of ALD. It has been treated as a sister technique of ALD and is specifically used for depositing organic and inorganic-organic hybrid materials [55]. MLD was first reported by Yoshimura et al. in 1991 [83] for growing a polyimide film. Thereafter, a variety of new polymeric films were reported, as recently reviewed by Meng [54,65]. Compared to inorganic coatings, polymeric films have benefits in their flexibility and elasticity due to their long chains of primary covalent bonds and secondary van der Waals bonds. In many cases, these reactions proceed without the need for nucleation enhancers, though organometallic compounds have also been used to improve reaction spontaneity and reduce nucleation delays [84]. ALD and MLD both recently have received increasing attention for their applications in new battery systems due to their distinguished capabilities for novel materials, and for the advancement of alkali metal anodes, as reviewed in this report.

## 3. Surface Modifications on Alkali Metals via ALD

ALD was first studied on alkali metal anodes in 2015 [30]. Al_2_O_3_ was deposited in a 14 nm layer on Li metal. Al_2_O_3_ has since been further developed by many different reports, including deposition on Na metal in 2016. On Li metal, additional oxides of ZrO_2_ and TiO_2_ have both been tested, as well as a ternary sulfide of Li_x_Al_y_S, LiF and AlF_3_, and a phosphate of Li_3_PO_4_. These coating materials and processes are summarized in Table 1 and Table 2. In these works, the alkali metal substrates have not been pretreated or functionalized before deposition. ALD has not yet been directly applied to K metal, but nanostructured K-metal hosts have been enabled by an ALD SnO_2_ coating.

### 3.1. ALD Coatings on Li Metal

#### 3.1.1. Oxide Coatings

The first report of ALD for surface engineering of Li metal was reported in 2015 by Kozen et al. [30]. In this work, a 14 nm-thick Al_2_O_3_ layer was deposited onto Li metal via a plasma-enhanced ALD process. Chemical resistance to air and electrolytes for Al_2_O_3_-coated and uncoated Li was studied qualitatively, demonstrating that the coating greatly improved the chemical stability of Li metal under these conditions (Figure 1a). After less than one minute of atmospheric exposure, the bare Li surface began to tarnish while the Al_2_O_3_-coated Li metal sustained in air (25 °C, 40% relative humidity (RH)) for 20 h. After immersion in a 1 M sulfur in dimethyoxylane (DME) solution for 7 d, the Al_2_O_3_-coated Li metal demonstrated remarkable stability with nearly no surface change. In contrast, the bare Li metal reacted dramatically with the solution, blackening over the course of the test due to reactions between the solution and electrode. Additionally, in Li-S full cell tests, after 100 charge/discharge cycles, an improved capacity retention was achieved, with ~90% for the Al_2_O_3_-coated Li metal anodes versus ~50% for the bare Li metal (Figure 1b). Kazyak et al. [85] revealed that an ultrathin (2 nm) Al_2_O_3_ coating still protected and improved the performance of Li metal. When cycled at 1 mA cm^−2^ with a fixed areal capacity of 0.25 mAh cm^−2^, the ultrathin coating resisted overpotential increase for nearly 800 charge/discharge cycles and allowed continuous cycling until 1200 charge/discharge cycles. They concluded that the ultrathin ALD Al_2_O_3_ coating helped contribute to uniform Li plating and suppression of dendritic growth to improve electrochemical performance and stability.

In a subsequent study, Lin et al. [96] revealed that ALD Al_2_O_3_ coatings offer the added benefit of smoothing out the Li surface. Using in situ atomic force microscopy (AFM), it was revealed that irregularities on the Li metal surface, such as ridges and cracks, served as preferential sites for solvent decomposition and SEI growth. After soaking bare and 10.5 nm Al_2_O_3_-coated Li metal in propylene carbonate (PC) for 12 h, there was no significant change in surface morphology or coating thickness for Al_2_O_3_-coated Li, while an SEI growth rate of 60 nm/h was estimated at ridge sites on the uncoated samples. In 2017, Chen et al. [87] studied the Al_2_O_3_ coating composition during cycling using X-ray photoelectron spectroscopy (XPS). Studying the elemental composition of the coating over time revealed minimal change before and after cycling, suggesting that the Al_2_O_3_ coating is chemically and electrochemically stable when in contact with the electrode and electrolytes. Using in situ transmission electron microscopy (TEM), it was also confirmed that Li ions diffuse through the coating and deposit into the underlying bulk Li, thereby resisting SEI formation due to deposition on top of the coating surface. In asymmetric Li-Cu cells at a current density of 1 mA cm^−2^, the Al_2_O_3_-coated Li maintained CE values as high as 98% beyond 180 charge/discharge cycles, more than two times longer than the cycling life of the bare Li.

In the most recent ALD for alkali metal anodes, Jin et al. [93] utilized a new two-step process to create a lithiated LiAlO_x_ coating layer without the need for a specific lithium precursor. Al_2_O_3_ was deposited by ALD at a chamber temperature of 120 °C, followed by a 30 min anneal at 180 °C while still in the ALD chamber. The additional high-temperature annealing time allowed Li from the bulk electrode to diffuse into the coating, enhancing its ionic conductivity. Time-of-flight secondary ion mass spectrometry (TOF-SIMS) depth profiling was used to study the elemental composition of the annealed coating layers. Compared to unannealed Al_2_O_3_, the annealed LiAlO_x_ demonstrated a high Li content at the outer surface of the coating, with intensity increasing towards the Li bulk. After cycling at 1 mA cm^−2^ to a fixed areal current of 1 mAh cm^−2^, TOF-SIMS was once again employed to study the evolution of the coating. The concentration of Li had greatly increased in post-cycling, and the profile was far more smoothly gradiated, still with increased composition near the bulk Li. When testing the electrochemical performance of LiAlO_x_-coated Li at 3 mA cm^−2^, two thicknesses of Li foil were used. On Li foils with thickness of 350 µm, the coated samples revealed stable overpotential of ~200 mV for more than 350 h. Comparatively, on Li foils with thickness of 40 µm, overpotential begins to increase just after 100 h. This change is due to the higher proportional loss of Li to side reactions and additional coating lithiation in the thin Li foil compared to the thick Li foil. This testing is important for the future of lithium metal anodes, as commercialized LMBs will likely use very thin layers of Li as their electrodes, and a coating must demonstrate stability even on electrodes with micro-scale thicknesses.

Compared to the widely investigated ALD Al_2_O_3_ coatings, ZrO_2_ coatings allow for Li anodes with higher rate capability. This may be due to the higher dielectric constant and carrier density properties of ZrO_2_ promoting Li ion transfer through the coating structure [97,98]. Alaboina et al. [88] deposited a ZrO_2_ nanofilm on Li metal via a plasma-enhanced ALD technique using tetrakis(dimethylamino)zirconium (TDMAZ) and water as precursors. The ZrO_2_-coated Li exhibited high resistance to corrosion under atmospheric conditions and excellent heat tolerance at temperatures near the melting point of Li metal (~180.5 °C [99]). Moreover, in symmetric Li-Li cells at 4 mA cm^−2^, the ZrO_2_-coated Li electrodes showed a more stable and lower overpotential cycling for over 100 charge/discharge cycles, while the bare Li-Li symmetric cells exhibited a continuously increasing overpotential profile. Electrochemical impedance (EIS) measurements at the first and 100th charge/discharge cycle revealed that the coated Li-Li cell initially showed a higher impedance than that of the bare Li-Li cell, but the maintained a significantly more stable impedance value after cycling (Figure 2a). To study the propagation of dendrites in the coated and uncoated cells, in situ optical microscopy images of the electrode/electrolyte interface were taken directly within 100 charge/discharge cycles. Dendritic growth was apparent in the bare cell after a single charge/discharge cycle and became more evident with more cycles. In contrast, the ZrO_2_-coated Li displayed flatter and more uniform dendrites (Figure 2b). These comparative results indicate that the ZrO_2_ ALD protection layer effectively passivates the electrode surface to prevent side reactions with electrolyte and mechanically suppress the growth of dendrites.

Li coated with a 5 nm (50 ALD cycles) TiO_2_ film has also demonstrated significant improvements in air stability, cycling stability, and rate capability [92]. After 8 h of air exposure, most of the TiO_2_-protected Li surface maintained its initial silver-white color with only some local areas displaying signs of tarnish while bare Li was completely corroded. Additionally, symmetric cells assembled with TiO_2_-coated Li could cycle stably with low voltage hysteresis for more than 1600 h at 1 mA cm^−2^, and more than 500 h at 3 and 10 mA cm^−2^. EIS revealed that the impedance of the TiO_2_ layer decreased with charge/discharge cycling, likely due to the transformation from amorphous TiO_2_ to a crystalline Li_2_Ti_2_O_4_ with a higher ionic conductivity [100,101]. In full-cell tests against an NMC622 cathode, the TiO_2_-coated Li improved cyclability over the uncoated Li, accounting for a capacity retention of 78.4% and 55.1% after 100 charge/discharge cycles, respectively. Ascribed to its high air tolerance, robust mechanical properties, and excellent electrochemical stability, TiO_2_ was regarded as one of the most promising of oxide coatings on Li metal.

#### 3.1.2. Fluoride Coatings

Inspired by a natural SEI component [102], its chemical inertness on lithium metal, and high electrochemical stability [103], in 2018, Chen et al. [89] deposited a homogeneous and stoichiometric LiF on Li metal with a GPC of 0.82–0.88 Å/cycle. In asymmetric Li-Cu cells, a stable CE as high as 99.5% for up to 170 charge/discharge cycles was obtained with the ALD LiF-coated Li, accounting for a 400% increase in cycling life over the uncoated Li. The improvement in electrochemical performance was attributed to two main coating characteristics: mechanical rigidity and interfacial passivation. The shear modulus of LiF was quantified at 58 GPa using nanoindentation, which is more than seven times the amount necessary to suppress lithium dendrites [104]. Moreover, the room-temperature ionic conductivity of LiF has been estimated at 10^−12^ S cm^−1^, ensuring a uniform ion flux that promotes planar electro-plating while also avoiding lithium deposition on the coating surface. They also found that the LiF coating provided a remarkable improvement of cycling lifespan even using an electrolyte volume of 5 μL, which is critical for practical commercialization of LMBs (Figure 3).

LiF is not the only fluoride-based coating that has shown improvement in electrochemical stability and performance for Li metal, as ALD AlF_3_ has been shown to dramatically improve performance in symmetric Li-Li and Li-S full cells [90]. In symmetric cells, the AlF_3_-coated Li showed a higher initial overpotential than that of the uncoated Li but cycled significantly longer before cell failure. The AlF_3_-coated Li was also highly effective in Li-S full cells, enabling a capacity two times higher than that of the uncoated cells. Energy-dispersive X-ray spectroscopy (EDS) was used to analyze coating evolution before and after charge/discharge cycling and revealed no significant reduction in the counts of aluminum and fluorine. This finding suggested that the coating was stable in electrolyte and allowed for lithium plating underneath the AlF_3_ coating.

#### 3.1.3. Phosphate Coatings

Li_3_PO_4_ has demonstrated a high elastic modulus and ionic conductivity in previous reports [105]. In 2021, Niu et al. [91] deposited amorphous Li_3_PO_4_ onto Li metal using lithium tert-butoxide and trimethyl phosphate (TMP). The resulting Li_3_PO_4_ coating can effectively suppress the growth of needle-like lithium dendrites and reduce side reactions between Li metal and electrolytes. Li_3_PO_4_-coated Li was also tested against bare Li for qualitative atmospheric tolerance. After atmospheric exposure for 6 h bare Li became completely black while there was barely surface change for the ALD Li_3_PO_4_-coated Li. This suggested a high resistance of the ALD Li_3_PO_4_ to O_2_, CO_2_, and H_2_O. Moreover, in symmetric Li-Li cells, the Li_3_PO_4_-protected Li could cycle stably for more than 1000 h at 0.5 and 1 mA cm^−2^, which was about 3 times that of bare Li. The researchers believed that the strong Li-P bonds in the coating might have helped provide a high elastic modulus for suppression of volume change and dendrite growth. Using a nanohardness tester, the elastic modulus of Li_3_PO_4_ was quantified at 10.1 GPa, which is believed to be sufficient for dendrite suppression according to the Newman and Monroe model [34]. In SEM observations, a more uniform and smoother top surface of Li_3_PO_4_ coated Li was found compared to the top surface of the bare Li after 30 charge/discharge cycles. Thus, the high elastic modulus, uniform, and dense Li_3_PO_4_ ALD coating has provided another possibility for stabilizing alkali metal electrodes.

#### 3.1.4. Sulfide Coatings

In 2016, an ionically conducting Li_x_Al_y_S coating with ionic conductivity of 2.5 × 10^−7^ S cm^−1^ at room temperature was coated on Li metal [86] with a GPC of 0.66 Å/cycle. In asymmetric Li-Cu cells, the CE of Li_x_Al_y_S was found to vary depending on the electrolyte. In a carbonate-based electrolyte of 1 M LiPF_6_ in 3:7 EC/EMC (EC = ethylene carbonate, EMC = ethyl methyl carbonate), the CE remained stably around 80% for more than 150 cycles, but a dramatic improvement to a CE of 97% beyond 700 cycles was noted in an ether electrolyte of 4 M LiTFSI (lithium bis(trifluroromethane)sulfonamide) in DME (1,2-dimethoxyethane) (Figure 4a). Top and cross-sectional SEM observations revealed that the Li deposited on the pristine Cu foil showed a narrow and needle-like structure, while the Li deposited on the Li_x_Al_y_S-coated Cu foil showed no obvious dendrite growth and the surface was uniform and dense (Figure 4b). This study demonstrated that the Li_x_Al_y_S coating could effectively stabilize the Li metal/electrolyte interface and inhibit the formation of Li dendritic structures. Ternary lithium-containing metal sulfides via ALD are a relatively new concept for alkali metal anode studies, but have showed among the most promising results in stabilizing Li metal for commercial use.

### 3.2. ALD Coatings on Na and K Metal

Over the past few decades, many efforts have been devoted to understanding the formation and growth mechanism of Li dendrites and developing various strategies for Li protection. Compared to Li metal, there is even more severe SEI and dendrite growth with Na and K metals due to their higher chemical reactivity [106]. Various strategies have been summarized to facilitate the efficient use of Na [107] /K [50] metal anodes. Since the melting point of alkali metals decreases moving down the group, ALD is among the most promising techniques in providing scalable surface coatings for Na and K electrodes owing to its low-temperature deposition compared to other coating methods.

In 2016, Luo et al. [94] utilized a plasma-enhanced ALD process using TMA and O_2_ plasma as precursors to deposit a 2.8 nm Al_2_O_3_ coating on Na metal. In symmetric Na-Na cells cycled at 0.25 mA cm^−2^, the Al_2_O_3_-coated Na began with a slightly higher initial overpotential, but remained stable beyond 450 h, while the bare Na exhibited a continually increasing overpotential (Figure 5a). To confirm that the Al_2_O_3_ coating layer was responsible for this change, EIS results were taken at the 1st, 100th, and 200th charge/discharge cycle. The initial impedance of the bare Na-Na cell was lower than that of the Al_2_O_3_-coated Na-Na cell. Upon cycling, however, the bare Na-Na cell showed a continual increase in impedance due to the formation of a native SEI, while the coated Na-Na cell showed an extremely stable impedance (Figure 5b). The rigid mechanical properties of Al_2_O_3_ [108] and facile sodiation [109] of the thin Al_2_O_3_ layer might have led to a more uniform Na stripping and plating processes. Moreover, top-view SEM observations of bare Na and Al_2_O_3_-protected Na after 450 charge/discharge cycles confirmed that the bare Na surface was almost entirely covered with three-dimensional (3D) Na dendrites, while the coated surface was smooth and planar (Figure 5c).

Al_2_O_3_ was also successfully coated on Na at 85 °C using a thermal ALD process by Zhao et al. [95] with a GPC of 0.14 nm/cycle. In Na-Na symmetric cells at a current density of 3 mA cm^−2^ in an ether electrolyte (1 M NaSO_3_CF_3_ in diethylene glycol dimethyl ether (DEGDME)), Na@25Al_2_O_3_, representing 25 ALD cycles of Al_2_O_3_ coated on Na, was chosen as the optimal coating thickness. A continual increase in voltage hysteresis was found in Na@10Al_2_O_3_, and a high impedance was noted in Na@50Al_2_O_3_ due to the thick coating layer. Na@25Al_2_O_3_ enabled excellent stripping/plating performance and an increased cycling life with almost no increase in overpotential after cycling over 70 h. After the first charge/discharge cycle, smooth, island-like Na with a diameter over 100 µm was observed on Na@25Al_2_O_3_, which was clearly distinguished from the moss-like and dendritic surface of the bare Na. In addition, after 10 charge/discharge cycles, the appearance of a rough surface littered with cracks can be observed on the bare Na surface. In contrast, the island-like Na on the surface of Na@25Al_2_O_3_ was still stable. These findings implied that the island-like Na structures were mechanically stable during cycling, compared to the moss-like and dendritic Na. The generation and growth of island-like Na might be attributed to homogeneous ion flux through the Al_2_O_3_ coating layer. Rutherford backscattering spectrometry (RBS) analysis after cycling revealed no Na peaks on the Al_2_O_3_-coated Na, demonstrating that the coating is stable in the electrolyte.

Sodium metal coated with Al_2_O_3_ can also be annealed to form sodiated NaAlO_x_, as also shown with LiAlO_x_ [93]. To optimize the process for Na, the deposition temperature was reduced to 65 °C, and annealing temperature reduced to 98 °C. From RBS analysis, it was noted that Na displayed superior diffusivity over Li, as peaks for Na were found as high as 40 nm above the electrode surface. In Na-Na symmetric cells assembled with 1 M NaOTf in DEGDME electrolyte, stability of the Na metal anode is greatly improved. While the thickness of the metal foil is unspecified, it was demonstrated that NaAlO_x_-coated Na can cycle stably for over 1000 h at 1 mA cm^−2^ and 3 mA cm^−2^ with overpotentials below ~40 mV.

Potassium metal batteries have not yet received as much research attention as Li and Na, and as such there have not yet been reports of ALD directly on K metal [110]. However, Zhao et al. [111] coated SnO_2_ on conductive porous carbon nanofibers (PCNFs) using ALD with Tetrakis(dimethylamino)tin (IV) and H_2_O as precursors. These composite mats, denoted PCNF@SnO_2_, served as a host for the K metal anode after a molten infusion of K at 250 °C, shown in Figure 6a. SEM and TEM observations verified no change in surface morphology of the PCNFs after the application of the SnO_2_ coating, implying a uniform and conformal deposition of SnO_2_ on the PCNF surface. Moreover, EDS mapping images further demonstrated that Sn and O elements were homogeneously distributed on the surface of PCNFs. In symmetric K-K cells cycled at 1 mA cm^−2^, the PCNF@SnO_2_-K anodes reveal low initial overpotential and a stable voltage hysteresis for more than 1700 h (Figure 6b). Comparatively, the bare samples began cycling with a much higher overpotential, and despite realizing decreased hysteresis over time, the cells suddenly failed just before 600 h. The difference in performance is attributed to the SnO_2_ coating layer providing a framework by which the K ions can diffuse into the PCNF substrate homogenously, while also suppressing K dendrite growth. In K-Cu asymmetric cells, bare K has a highly fluctuating CE due to uneven K plating and stripping, but the PCNF@SnO_2_-K composite anode delivers a high and stable CE with average of 98.3% over 150 charge/discharge cycles (Figure 6c).

Due to their reduced energy storage capability and chemical stability, Na and K metal have not yet been as extensively studied as Li metal. Despite this, further research into sodium and potassium metal is critical to achieving sustainable electrochemical energy storage, and ALD is likely to play a vital role in the production of future NMBs and KMBs.

## 4. Surface Modifications on Alkali Metals via MLD

Compared to dense and stiff inorganic ALD coatings, the reduced density and increased porosity of MLD coatings are expected to accommodate alkali metal volume change. Further, higher tuneability of coating structure due to the integration of organic bonds in MLD coatings offers coatings with a wider range of effective mechanical, chemical, and electrical properties. MLD has not yet been as extensively studied as ALD for alkali metal anodes, but has already shown great performance improvements and novel materials for stabilization of alkali metals. The first report for MLD on alkali metal anodes was in 2017 when aluminum ethylene glycol (AlEG) was deposited onto sodium metal [112]. Since then, Li metal has been coated with AlEG, zirconium ethylene glycol (ZrEG), lithium glycerol (LiGL), and polyurea (PU), as summarized in Table 3 and Table 4. As with ALD, there has not yet been a report for MLD on K metal anodes.

### 4.1. MLD Coatings on Li Metal

#### 4.1.1. Alucone Coatings

As an analogue of ALD, MLD can not only deposit inorganic-organic hybrid thin films, but also enable the growth of purely organic thin films, which demonstrate many advantages in coating uniform and conformal thin films even on high-aspect-ratio 3D substrates [119,120]. The addition of new flexible bonding structures provides precise and tailorable control over film thickness and chemical composition at the nano scale [121]. Moreover, the introduction of C-C and C-O bonds into MLD coatings is expected to offer a new solution to accommodate/suppress dendrite growth and the huge volume expansion of alkali metal anodes [112,122]. Many inorganic-organic hybrid MLD coatings based on metals and organic alcohols yield metal alkoxide films referred to as “metal-cones”, as summarized by George et al. [123]. Aluminum ethylene glycol is an “alucone” that was first grown via MLD by Dameron et al. [121] using TMA and ethylene glycol (EG) as precursors. Benefiting from low-cost and readily available precursors, as well as a wide deposition temperature range, the researchers demonstrated that AlEG can be effectively coated between 85 and 175 °C with a GPC of 4.0 Å/cycle at low temperature and 0.4 Å/cycle at high temperature. Further contributing to its popularity, MLD AlEG enables surface coatings at low-temperatures for stabilizing sodium metal anodes [112,118] and high-temperature coatings for lithium metal [113,114].

In 2018, Chen et al. [113] deposited 30 and 60 MLD cycles (corresponding to ~3 nm and ~6 nm, respectively) of AlEG on Li metal at 150 °C. These coated samples were tested in symmetric Li-Li cells using 5, 10, and 20 μL of a carbonate-based electrolyte (1 M LiPF_6_ in 3:7 EC/EMC). Compared to uncoated Li-Li cells, the AlEG-coated Li-Li cells enabled a consistently reduced overpotential when tested at 1 mA cm^−2^ with an areal capacity of 1 mAh cm^−2^. From these tests, the 6 nm coating was determined to perform more optimally than the 3 nm coating. The AlEG-coated Li enabled lower overpotential and a more stable voltage hysteresis than the bare Li. Further, the lifespan of the bare Li cells decreased with leaner electrolyte volumes, but this phenomenon is far less marked in AlEG-coated cells due to reduced electrolyte consumption. AlEG-coated Li was also tested in Li-S full cells, where it was observed that Li-S cells using the AlEG-coated Li maintained their initial voltage profile significantly longer than uncoated Li. The 39.5% increase in initial capacity of AlEG-protected Li-S compared to traditional Li-S marks a significant improvement in the viability of next-generation Li-S with protected lithium metal anodes (Figure 7a).

AlEG was also directly deposited on Li surface by Zhao et al. at 120 °C [114], having a GPC of 0.5 nm/cycle. Li electrodes with AlEG coatings of 5, 10, 25, and 50 MLD cycles were assembled into symmetric Li-Li cells with a carbonate electrolyte (1 M LiPF_6_ in EC:DEC:EMC of 1:1:1 volume ratio) (Figure 7b). At 5 mA cm^−2^, the AlEG-Li coated Li-Li cells enabled a significantly improved cyclability over that of the uncoated cells. The optimal coating thickness was 10 MLD cycles. After 10 charge/discharge cycles, SEM images were taken of coated and uncoated samples. The uncoated Li displayed a highly porous and irregular surface, owing to multiple adverse reactions at the electrolyte interface, while the AlEG-coated sample maintained a more uniform and homogenous surface morphology. These findings suggest that not only does AlEG possess sufficient chemical resistance to ether electrolyte but also mechanical rigidity to reduce dendrite growth and accommodate volume change. The researchers concluded that AlEG is superior to Al_2_O_3_ as a surface coating for Li metal in both carbonate and ether electrolytes, attributed to the higher electrochemical stability, mechanical flexibility, and Li^+^ conductivity of AlEG.

Additionally, MLD AlEG has been used to stabilize the interface between Li metal and solid-state electrolytes (SSEs) [115]. By applying AlEG on Li metal in thicknesses of 10, 30, and 50 MLD cycles (denoted as Li@10Alucone, Li@30Alucone, and Li@50Alucone), the electrochemical performance of symmetric Li/Li_10_SnP_2_S_12_/Li cells at 0.1 mA cm^−2^ was greatly improved. Li@30Alucone was ideal for this application, as Li@10Alucone was insufficient to provide steady overpotential free from growth, and Li@50Alucone revealed incredibly high overpotential due to sluggish reaction kinetics in the thick coating layer (Figure 8a–c). This was confirmed further by time-dependent EIS of all coating thicknesses. The EIS spectra revealed that the initial impedance of all coated cells was higher than the bare cell, but the Li@30Alucone demonstrated the lowest impedance of the coated cells with the lowest growth within 24 h cycling. Accordingly, the overpotentials of all the coated cells started higher than bare Li-Li cell, but the Li@30Alucone quickly demonstrated the most stable overpotential with the least overpotential increase after 150 h. From the electrochemical performance and EIS results, it was concluded that the AlEG coating can effectively suppress the interfacial reactions between Li and Li_10_SnP_2_S_12_. This was verified by XPS analysis of the AlEG-coated and uncoated Li electrodes after 25 charge/discharge cycles. For the uncoated Li, it was found that Sn^4+^ from the electrolyte was reduced to Sn^2+^ due to irreversible reactions at the electrode/electrolyte interface. In contrast, the Li@30Alucone revealed no change in the intensity of the Sn 3d peak, suggesting a successful suppression of interfacial reactions. MLD interface engineering for SSEs also reveals advantages beyond symmetric cells, as full cells with LiCoO_2_ cathodes were tested at 55 °C and 0.1 C (1 C = 140 mA g^−1^) to show significantly higher capacity retention and Coulombic efficiency in AlEG-coated all-solid-state LMBs after 150 cycles (Figure 8d). Therefore, the conformal, flexible alucone coatings can effectively reduce undesirable side reactions and promote a more uniform lithium diffusion during charge/discharge, which offers a promising avenue toward high-performance and stable solid-state LMBs.

#### 4.1.2. Zircone Coatings

Another family of inorganic-organic hybrid materials that have been tested on alkali metals are zircones. Zircone MLD in the form of zirconium ethylene glycol (ZrEG) was first fabricated using alternating pulses of zirconium tert-butoxide (ZTB) and EG at 145 °C with a GPC of 0.78 Å/cycle [124]. In 2018, Adair et al. [117] deposited ZrEG on lithium metal at 130 °C with a GPC of ~1.5 Å/cycle. Coating thicknesses of 10, 25, and 50 MLD cycles were tested, and characterization of the coating was conducted using TOF-SIMS. ZrCOH- and CH- were detected in the deposited thin film, confirming the presence of zirconium-rich and organic species. ZrEG-coated Li foil was then assembled into symmetric Li-Li coin cells and cycled at 3 and 5 mA cm^−2^. In all cases, the 25 MLD cycled coating (Li@25ZrEG) performed most optimally and realized higher overpotential stability when compared with bare samples. It is believed that the 25 MLD cycles of ZrEG are sufficient to provide a complete and uniform coating of the Li metal surface while also minimizing the interfacial impedance.

After 30 charge/discharge cycles, cells were disassembled and observed using SEM. The uncoated Li had a jagged and irregular surface with deep cracks, while the ZrEG-coated Li maintained a much more uniform surface with only small hairline cracks (Figure 9a). This can be attributed to successful deposition of lithium underneath the coating during electroplating, as well as the beneficial mechanical strength of ZrEG in suppressing dendrites and volume change. The evolution of ZrEG was also studied using X-ray absorption near edge structure (XANES) scanning. The XANES incident X-rays were able to penetrate through the lithium metal and analyze the electronic structure of the ZrEG coating during cycling. Increased intensity during the first plating sub-cycle likely demonstrates that Li can diffuse into the ZrEG matrix, lithiating the coating over time and improving its conductivity. Of important note are the benefits of Li@25ZrEG in improving the air stability of the Li foils. When left in atmospheric conditions, ZrEG-coated Li resisted the effects of tarnishing due to reaction with oxygen and water vapor for several hours. To further analyze the chemical stability of ZrEG in oxygen-rich environments, bare Li and Li@25ZrEG were assembled into Li-O_2_ batteries. Cycling ZrEG-Li-O_2_ reveals a far more stable voltage window with significantly longer cycle life compared to bare samples (Figure 9b). Zircones may pose different solutions compared to alucones on alkali metals due to their differing chemical and mechanical properties. More studies are necessary to gain an improved understanding of how different metallic or organic components in otherwise similar coatings reveal different advantages for certain applications.

#### 4.1.3. Lithicone Coatings

Lithicones of lithium propane dioxide (LPDO) were first proposed by Wang et al. [125] in 2020 using lithium tert-butoxide (LTB) and 1,3-propanediol (PDO) as precursors between 150 and 200 °C with a GPC between 0.23 and 0.15 Å/cycle. Other studies with lithium ethylene glycol (LiEG) [126], revealed high a high ionic conductivity of 3.65 × 10^−8^ S cm^−1^ at room temperature. The most promising work in both LMB and MLD research to date has come in the form of lithium glycerol (LiGL) proposed in 2021 by Meng et al. [57]. In this study, LiEG, LiHQ (lithium hydroquinone), and LiGL were deposited onto lithium metal anodes. All lithicones were deposited at 150 °C, with an initial dose of LTB followed by a dose of the chosen organic molecule (EG, HQ, GL) in an alternating pattern. LiGL demonstrated the highest growth rate yet seen in MLD coatings with a GPC of 2.7 nm/cycle while also promoting cross-linked growth between layers due to the high number and asymmetric layout of hydroxyl terminations in the glycerol molecule. It was noted that the growth rate of LiGL begins slowly but increases over time until achieving a linear growth rate, potentially due to nucleation delays in the initial cycles. The effects of LiGL were measured in symmetric Li-Li cells using ether electrolyte (1 M LiTFSI in 1:1 DOL/DME) without electrolyte additives. Coating thicknesses of 10, 15, 20, 60, and 90 MLD cycles were analyzed at 2, 5, and 7.5 mA cm^−2^ with a fixed areal capacity of 1 mAh cm^−2^. At 2 mA cm^−2^, all samples began with similar overpotentials, but the bare Li-Li cell quickly saw uncontrolled overpotential increase near 600 charge/discharge cycles. Comparatively, the LiGL coatings realized significantly higher cycling stability, with thicker coatings up to 90 MLD cycles demonstrating lower and more stable overpotentials (Figure 10a). Higher rate testing revealed similar results, with testing at 5 mA cm^−2^ and 60 MLD cycles realizing stable cycling performance of ~200 mV overpotential after 13,500 charge/discharge cycles (Figure 10b). The elastic modulus of LiGL was obtained using atomic force microscopy (AFM). Interestingly, the elastic modulus decreased as number of coating cycles increased, suggesting that the coating becomes more malleable in thicker layers. In post-charge/discharge-cycling SEM observations of LiGL-coated and bare Li, dendrite-like structures can be found on the bare and thinly coated Li electrodes while a smoother and more uniform surface can be observed in the LiGL-60 electrode, suggesting that the LiGL film is sufficient to inhibit Li dendritic growth and protect the ether electrolyte from decomposition. Though only one lithicone has yet been intensely tested on alkali metals, the future for lithicones appears bright as LiGL has currently displayed the highest cycling stability yet reported in any literature for LMBs. A sound growth mechanism has not yet been proposed for lithicones, as the layer-by-layer growth scheme seen in ALD and MLD coatings is not conducive with alkali frameworks owing to their single valence electron. For this reason, it may be assumed that the growth of alkali-rich ALD and MLD coatings follows a more complex growth mechanism than multivalent metallic coatings. This change in film growth and propagation may reveal a more ideal combination of ionic conductivity, chemical stability, and mechanical stiffness for alkali metal anodes. It is for these reasons that lithicones represent one of the least understood, but most promising avenues for development of commercial LMBs.

#### 4.1.4. Purely Organic Coatings

Metalcones are the most intensively studied family of MLD coating materials on alkali metal anodes, but MLD is also capable of producing fully organic coatings. Purely organic coatings are of particular research interest because their high concentration of covalent C-C and C-O bonds may provide different chemical and mechanical properties to accommodate the volume change of alkali metal anodes. One purely organic coating that can be grown by MLD is polyurea (PU). Growth of nanoscale MLD-PU films was first realized in 2010 by Loscutoff et al. [127] using 1,2 phenylene diisocyanate (PDIC) and ethylenediamine (ED) with a GPC of 0.65 nm/cycle. Of important note are the low-temperature deposition potential of PU coatings, as well as the lack of byproduct generation. MLD PU-coating was first applied to lithium metal in 2018 by Sun et al. at a very low temperature of 65 °C [116] after simulation results proposed that the nitrogen-containing polar groups may help regulate Li-ion flux and lead to a more uniform Li plating morphology [128]. Different coating thicknesses of 5, 10, 25, and 50 cycles were tested to find the most optimal coating thickness for this PU-coating. TOF-SIMS was used to verify the presence of of C_x_H^−^, CN^−^, and NCO^−^ before and after Cs^+^ sputtering, confirming deposition of organic and nitrogen-rich groups on the Li surface. High intensities of organic macrostructures are seen before sputtering, with a notable decrease in concentration after 70 s of sputtering. This change in intensity verifies the organic coating was deposited on the surface and did not influence the bulk electrode structure.

In Li-Li symmetric cells cycled at 1 and 3 mA cm^−2^ to capacity of 1 mAh cm^−2^ (Figure 11a), the PU-coated Li realized more stable voltage hysteresis, particularly at high rates and capacity, over uncoated Li. At 3 mA cm^−2^, the bare samples reached overpotential above 500 mV before cell failure at 80 h, while the PU-coated sample shows a more stable overpotential growth of <200 mV even after 200 h. Low rate and high-capacity tests of 1 mA cm^−2^ and 2 mAh cm^−2^ further demonstrate the protective effects of the PU-coating, as the coated sample maintains a relatively stable overpotential profile beyond 400 h (Figure 11b). XPS analysis of the coated samples before and after 20 charge/discharge cycles found strong and persistent nitrogen and carbon peaks, verifying that the coating structure remains intact and stable during cycling. Further, a high LiF content was found in the SEI formed on the coating surface, which may result from reactions between the -NH groups in the PU coating and PF_6_^−^ electrolyte ions. Because LiF has an ultra-high shear modulus [89], it can be further beneficial to stabilize the SEI and suppress Li dendrite growth.

Moreover, the effects of bare Li full cells and PU-coated Li/LFP full cells were tested for their rate capability under current densities ranging from C/10 to 4 C (1 C = 170 mA g^−1^). The initial performances of PU-coated and uncoated cells were very similar, but at higher rates of 2 and 4 C, the PU-coated Li realized much higher capacities, as well as improved capacity retention after the samples returned to low-rate C/10 charging rate. This difference is believed to result from the highly stable artificial PU coating acting as a stable SEI for the Li, while the naturally occurring SEI on the uncoated Li hindered kinetics while forming dendrites and dead Li. While the PU coating has shown high electrochemical performance and stability, purely organic coatings have still not been extensively studied on alkali metal anodes, and a more detailed understanding of their operation may serve to fabricate stable and safe Li metal anodes.

### 4.2. MLD Coatings on Na Metal

Inspired by their previous advancements in alucone-coated carbon/sulfur cathodes [129,130], the first research in MLD on alkali metal anodes was conducted by Zhao et al. in 2017 [112]. AlEG was deposited on sodium metal via MLD at 85 °C. Coating thicknesses of 10, 25, and 40 MLD AlEG cycles were tested in Na-Na symmetric cells in carbonate electrolyte (1 M NaPF_6_ in 1:1 EC/PC) at 1 and 3 mA cm^−2^. From these tests, it was found that 25 MLD cycles of AlEG (Na@25Alucone) were optimal for protection and stability of the Na metal anode. Na@10Alucone did not provide sufficient protection from dendritic growth and led to sudden cell failure, while the Na@40Alucone displayed initially promising results but was prone to sudden and irregular voltage spikes before cell failure. Bare Na and Na@25Alucone samples began with similar overpotentials until 150 cycling hours, when the instability of the uncoated Na caused drastic overpotential spikes (Figure 12a). Investigation into the surface chemistries of bare Na and Na@25Alucone was conducted with XPS after 10 charge-discharge cycles. Lower intensities of 1 s orbitals for F, O, and C were found on bare Na compared to Na@25Alucone. This reduction in intensity on bare Na suggests that the electrolyte was consumed due to SEI formation. Moreover, the SEI on Na@25Alucone showed higher concentrations of beneficial NaF and Na_2_O, indicating trace concentrations of stable SEI components formed on the AlEG surface during cycling. Surface morphologies of the coated and uncoated samples after 10 charge-discharges were analyzed using SEM. The AlEG-coated Na showed no mossy or dendritic growth, as well as showing a robust connection between the electrode and coating layer through cross-sectional imaging. The uncoated Na surface exhibited irregular moss-like and 3D sphere-like Na dendrites, and the surface roughness increased with continuous cycling. Further, RBS measurements before and after 10 charge-discharge cycles revealed an increase in Na composition of the coating after cycling, suggesting that Na integrates into the coating structure over time while also remaining stably at the coating/electrode interface.

The effects of MLD AlEG not only benefit liquid-state batteries, but also next-generation solid-state Na batteries [118]. The same AlEG coating as the previous study was investigated as an interfacial contact stabilizer between Na metal and Na_3_SbS_4_ and Na_3_PS_4_ solid-state electrolytes. Coating thicknesses of 50, 150, and 300 MLD cycles (Na@mld50C, Na@mld150C, and Na@mld300C) were assembled into Na-Na symmetric cells with either Na_3_SbS_4_ or Na_3_PS_4_ as the electrolyte. Analysis of the Na@mld150C using ToF-SIMS found few Na^−^ signals and uniformly dispersed C_2_Al^−^ and AlO^−^, indicating the Na surface was homogeneously coated and fully covered by the AlEG coating. Cycled at 0.1 mA cm^−2^ to a cutoff capacity of 0.1 mAh cm^−2^, Na@mld150C was found to perform most optimally. Na@mld150C maintained a more stable overpotential compared to the uncoated samples (Figure 11b). Overpotential of Na@mld150C stabilized at 450 mV, while bare Na realized higher overpotential greater than 1000 mV before cell failure. This difference may be ascribed to the successful reduction of reactions at the electrode/electrolyte interface, mechanical suppression of volume change from the AlEG coating, and a more uniform electrode/SSE interface. Interestingly, Na@mld300C did not improve on the bare sample, and the cell failed prematurely, potentially due to the increased density of AlEG when deposited at significantly thick layers. Increasing the density may offer beneficial mechanical properties, but greatly increases the interfacial impedance of Na^+^ transport. After 10 charge/discharge cycles, SEM imaging of uncoated Na revealed a highly irregular and rough surface characteristic of dendrite growth, while the AlEG-coated Na showed uniform and smooth surface. Na@mld150C was also investigated in all-solid-state cells with against a titanium sulfide (TiS_2_) cathode. At room temperature, cells with Na@mld150C revealed an initial specific capacity of 200 mA g^−1^ at a current density of 0.11 mA cm^−2^ and a capacity retention of 70% after 30 charge/discharge cycles. In contrast, bare Na cells realized sudden capacity loss and total cell failure after 20 charge/discharge cycles. It is demonstrated that MLD coatings on alkali anodes may adopt more than one crucial role in the next generation of metal-based batteries, acting as not only an artificial SEI for the anode, but also an interfacial buffer to ensure uniform contact between the solid anode and solid electrolyte. Despite this potential benefit, there is still a need for more stable solid electrolytes and further research into optimal coating materials.

## 5. Dual Layered Coatings on Li Metal via ALD and MLD

Dual-layered coatings aim to achieve an ideal balance between the properties of ALD and MLD coatings. A dual-layered ALD/MLD coating was first reported in 2019 using Al_2_O_3_ and AlEG on Li metal [131]. In 2021, another ALD/MLD hybrid technique was used on Li metal to deposit a hybrid polyurea (HPU) gradiated with zinc [132]. Thus far, only two reports have combined the properties of ALD and MLD to create a dual-layered material, as summarized in Table 5, and the potential benefit of these coatings has not yet been explored on Na or K metal.

As ALD and MLD offer individual benefits and drawbacks, recent studies have created a new class of hybrid materials in the form of dual-layered ALD and MLD coatings. Using a dual-layered technique, the conventionally superior mechanical rigidity and chemical inertness of many ALD coatings can blend with improved ionic conductivity and high flexibility of MLD coatings. The potential of using both MLD and ALD to stabilize the lithium metal anode was first studied in 2019 by Zhao et al. [131] using an inorganic Al_2_O_3_ ALD coating as inner protective layer and organic AlEG MLD coating as outer passive layer. This coating layer order was inspired by natural SEI dual protective layers in which the dense inner Al_2_O_3_ layer would resist stray electrons during Li^+^ transport, while the more flexible outer layer of AlEG would conform to outward volume expansion of Li metal. Coating deflection vs. applied force was studied using AFM, in which it was found that a higher magnitude of deflection was present with the organic outer layer than an inorganic outer layer, further validating the expected behavior of the dual layered coating approach.

Electrodes coated with 50MLD/50ALD/Li (organic outer layer and inorganic inner layer), 50ALD/50MLD/Li (inorganic outer layer and organic inner layer), 50ALD/Li, 50MLD/Li, and bare Li were assembled into symmetric Li-Li cells with carbonate or ether electrolyte. Both 50MLD/50ALD and 50ALD/50MLD samples demonstrated the most stable cycling performance in the carbonate electrolyte (1 M LiPF_6_ in 1:1:1 EC/DEC/dimethyl carbonate (DMC) with 10% fluoroethylene carbonate (FEC)), with the 50ALD/50MLD sample failing after 800 hrs. The 50MLD/50ALD/Li presented excellent stability with low overpotential of ~130 mV over 700 h. Compared with pristine Li, the cycling lifetime of the 50ALD/Li and 50 MLD/Li is also improved, but not to the same extent as the dual-layered coatings. Coated cells in ether electrolyte realized larger, but more stable overpotential in the charge–discharge process, with the 50MLD/50ALD still offering the most ideal cycling performance. At 5 mA cm^−2^ and 1 mAh cm^−2^, the 50MLD/50ALD symmetric cell displays overpotential of ~200 mV after 1300 hrs of cycling, while the 50ALD/50MLD displays higher ~250 mV after 1300 h (Figure 13a). Similarly, successful results were also found in low-rate and high-capacity tests. The 50MLD/50ALD/Li electrodes were also tested in full-cell Li-LFP, Li-S, and Li-O_2_ batteries, demonstrating that the optimized dual protective layer of 50MLD/50ALD can remarkably improve cycling stability, lifetime, and capacity of those batteries (Figure 13b).

In 2021, an advancement on previous polyurea study using MLD was made by Sun et al. [132]. Li metal was coated by a nanoscale polyurea film with an inorganic zinc framework to create a gradiently zincated polyurea structure. This dual layered material was created using diethyl zinc (DEZ) as the zinc-rich precursor, with ethylenediamine (ED) and 1,4-phenylene diisocyanate (PDIC) used as the polyurea precursors. The PDIC was heated to 90 °C to increase its vapor pressure, but the deposition was carried out at 65 °C. To produce this hybrid polyurea (HPU) coating, DEZ was initially dosed on the surface, followed by ED, PDIC, and a second dose of ED. Two different gradient directions were tested with ‘gradient coating’ referring to increased zinc concentration near the Li foil (HPU as inner layer and PU as outer layer) and ‘reverse gradient coating’ referring to increased zinc concentration near the electrolyte interface (PU as inner layer and HPU as outer layer). Gradient coatings (Li@Gr), reverse gradient coatings (Li@RGr), nongradiated coatings (Li@NonG), and standard polyurea (Li@PU) were applied to lithium foil and assembled into symmetric Li-Li cells. These cells were cycled at 4 and 6 mA cm^−2^. Of the different coatings, the dual-layered Li@Gr10 displayed the highest stability, with the Li@RGr10 producing the second-best results. These improvements over standard PU coatings can be ascribed to that the introduction of inorganic Zn components facilitating a uniform Li ion flux and regulating the plating/stripping behavior owing to the increase in amount of lithophilic sites. Li@Gr10 is presumed to be superior to Li@RGr due to a more optimal layout of coating chemical properties. In Li@Gr10, the outer PU layer forms an electrically insulating electrolyte interface, while the inwardly zincated structure increases the number of lithophilic sites closer to the Li electrode. In Li@RGr10, the lithophilic outer surface may have higher spontaneity for reactions with electrolytic ions. Li@Gr10 has also displayed some of the most promising results in Li-O_2_ full cells, with stable cycling over 1500 h at a capacity of 0.2 mAh cm^−2^.

SEI-like dual-layer coating techniques show high potential for improving the performance of alkali metal anodes. Similar phenomena has been noted in other coating techniques, as more diversified coating materials offer variable electrochemical and mechanical properties, thus allowing for a more robust and adaptive coating layer [133]. In both cases of dual-layered coating materials, high performance has been realized with dense inner layers and open organic outer layers, offering a helpful guide for future dual-layered studies. The dual layered technique may present a feasible pathway for safe and stable high-energy-density alkali metal batteries as further studies produce more surface coatings on alkali metal anodes.

## 6. Conclusions and Outlook

AMBs (lithium, sodium, and potassium metal) all suffer from stability and safety issues as they are cycled due to SEI buildup, volume change, and dendrite growth. Many different methods have been proposed to address these concerns, but among those with the easiest scalability and most tuneability are ALD and MLD. ALD and MLD both benefit from nanoscale control, low-temperature deposition, and highly conformal coatings on substrates of any geometry, enhancing their potential as surface coating techniques for alkali metal anodes. Furthermore, the vapor-phase and self-limiting coating construction offered by these techniques generates a high degree of scalability compared with other coating methods such as electrospinning or drip casting. These coatings provide interfacial passivation layers to inhibit SEI formation while also suppressing dendrite growth and volume change through their mechanical moduli. As fledgling areas of study, ALD/MLD have not yet been as extensively studied as some other stabilization methods but have already seen great improvements and developments towards safe alkali metal anodes. This work comprehensively summarizes recent progress on ALD/MLD for stabilizing alkali metal anodes including 12 different materials over 4 material categories, as summarized in Table 1, Table 2, Table 3, Table 4 and Table 5. Alkali metals represent an important advancement for the next generation of energy storage, but there are still many problems with coated alkali metals that need to be addressed. While Li has been the most studied of the alkali metal anodes, there is still not a perfect solution using ALD/MLD to properly stabilize LMBs. Further, the differences in melting point and chemical reactivity of Na and K require their own unique efforts to solve. Therefore, new precursors and novel ALD/MLD processes for coating all alkali metals are urgently needed to achieve precisely controllable protective films with low-temperature deposition, high mechanical properties, robust chemical stability, and superionic conductivity.

The most promising studies for ALD/MLD on alkali metals have come from dual-layered coatings and alkali-rich coatings. From dual-layered coatings, it has been shown that optimized layouts with dense inner layers and flexible outer layers provide excellent support to suppress dendrite growth while also accommodating volume change. Further, the diversified coating composition may also provide enhanced chemical support through variable chemical and electrochemical resistance. Alkali-rich coatings, while not yet as well understood as some other coating types, have demonstrated the most successful results to date for Li metal. Additional research into the growth scheme and in situ behavior of alkali rich coatings may provide new research channels by which they can be further optimized. These coating types will likely be the first groups to successfully mitigate the safety risks associated with alkali metal anodes and bring forward a new generation of high energy-density batteries. Still, more development is required to assess the cost-efficacy of applying ALD and MLD coatings at the commercial scale, while also improving knowledge of ideal coating configurations in different operational environments.

ALD coatings of Li_x_Al_y_S and TiO_2_ have produced highly stable Li metal anodes, and may be promising for LMBs in carbonate electrolytes due to their relative ease of deposition and chemical inertness. Thus far, carbonate electrolytes have proven to be a difficult challenge for organic coatings, and as such the future of high-voltage AMBs with alkali metal anodes may see higher improvement from inorganic ALD coatings compared to organic MLD coatings. Surface engineering through MLD has shown the highest potential for lower voltage next-generation LMBs such as Li-S and Li-O_2_ batteries using ether-based electrolyte. As new coating materials and deposition techniques are developed, a better understanding of the practical use for AMBs enabled by ALD and MLD will form.

Because the alkali metals characteristically react with O_2_, H_2_O, and CO_2_, it is still challenging to handle and characterize pristine and ALD/MLD coated alkali metals in the atmosphere. To this end, it will be of great benefit to implement in situ characterization techniques such as quartz crystal microbalance (QCM), Fourier-transform infrared spectroscopy (FTIR), Raman spectroscopy, AFM, XPS, SEM, TEM, etc. [134]. Additionally, although it has been qualitatively proven that ALD/MLD coated alkali metal electrodes effectively improve the electrochemical performances in various cells, there is still minimal quantified information about mechanical, chemical, and electrical properties of these ALD/MLD coating layers. Quantifying these properties will allow for trends and patterns between coating composition and property to form, further expediting the development of ALD/MLD coatings for alkali metal anodes.

In summary, this review has covered emerging ALD/MLD coatings for rechargeable AMBs, and the merits for and against each coating material were weighed. Deposition parameters, characterization, and electrochemical performance were reviewed to give readers the most pertinent and useful information regarding the properties of different applied coatings. It is believed that ALD and MLD coatings are among the most powerful, scalable, and tailorable techniques yet applied to alkali metal anodes. With this review, we hope to motivate more research into new ALD/MLD coating materials and characterization methods building on the success of previous coatings to bring forward safe alkali metal electrodes ready for use in the next generation of electrochemical energy storage.

## Data Availability

Not applicable.

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
