# Peer review of "Atomic and Molecular Layer Deposition as Surface Engineering Techniques for Emerging Alkali Metal Rechargeable Batteries"

_molecules, 2022, doi:10.3390/molecules27196170_

Round 1

Reviewer 1 Report

The reviewed article is part of a mainstream body of work helping to develop new technologies. At a time of striving to reduce fossil fuels, all research into different types of cells and batteries is badly needed. There is an urgent need for the development of new battery technologies and materials to further improve energy density, extend life, improve safety and reduce costs.

The alkaline metals and their application methods described in this paper can play a huge role in the development of next-generation battery technology.

New precursors and novel ALD / MLD processes for coating all alkali metals can be used to obtain carefully controlled protective films with low deposition temperatures, high mechanical properties, robust chemical stability and super ionic conductivity.

The article definitely fulfils the requirements of a scientific text. Formally, it is flawless. The structure of the reviewed article is thoughtful and clear. Therefore, I can recommend this paper as suitable for publication.

Author Response

Reviewer: 1                                                     

General Comments:

The reviewed article is part of a mainstream body of work helping to develop new technologies. At a time of striving to reduce fossil fuels, all research into different types of cells and batteries is badly needed. There is an urgent need for the development of new battery technologies and materials to further improve energy density, extend life, improve safety and reduce costs.

The alkaline metals and their application methods described in this paper can play a huge role in the development of next-generation battery technology.

New precursors and novel ALD / MLD processes for coating all alkali metals can be used to obtain carefully controlled protective films with low deposition temperatures, high mechanical properties, robust chemical stability and super ionic conductivity.

The article definitely fulfils the requirements of a scientific text. Formally, it is flawless. The structure of the reviewed article is thoughtful and clear. Therefore, I can recommend this paper as suitable for publication.

Reply: Thank you so much for the many positive comments on our work and the recommendation for publishing.

Comments:

Reply: There are no correctional comments to be addressed from this reviewer.

Reviewer 2 Report

The authors have reviewed in detail the application of ALD and MLD or their combination ALD/MLD technologies in surface engineering or modification/protection of electrode materials used in Li, Na and K batteries, highlighting the advantages of using such deposition technologies, specially at nanoscale.

The structure of the manuscript and content is very interesting but some minor improvements must be done in order to reach the desired level of quality. Thus, please pay attention to the following comments and suggestions:

Introduction is well suitable to the presented work. The authors introduced the problematic is clear as well as solution, in particular the SEI formation as well the swelling effect denoted as volume change. The strategies to enhance the electrodes activity are well described. The figures are appropriate, however, considering the Scheme 2 which illustrates the typical and well known Al2O3 ALD process, the authors should add also a new scheme illustrating the alucone formation process from the reaction of TMA + EG (replacing the H2O), which means the inorganic-organic film formation it will enrich the introduction and it will help the readers that are new to the ALD/MLD topic.

Since the Al-, Zn-, Ti-organic hybrid materials derived from the three widely used metal precursors from the ALD technology. The data presented and summarized in the Tables is well organized and clear. The legend in “Table 3.

Summary of MLD Surface Coating Li Metal” should be considered to be changed for ALD/MLD (inorganic + organic), instead of MLD (organic + organic), however PU No. 4 is in fact a MLD process (ethylenediamine + 1,4-phenylene diisocyanate) and eventually the titles/subtitles of the main text.

It is clear that the Al-based processes are extensively studied in this application that can be directly related to the TMA chemical reactivity even at low temperature deposition. Interestingly, the authors do not mention any nucleation problems or nucleation delays in the Li, Na, and K materials towards the film formation.

Prior to the deposition processes are the samples pre-treated or functionalized? Or the layers are deposited directly? Is this information provided in the reviewed literature?

Finally, the Conclusions are written in detail with numeric data, highlighting the best results, including characterization methodologies to better understanding the effect of the ALD/MLD coatings on the electrodes performance. I recommend the paper for publication, however, there are some concerns, comments and suggestion should be addressed before publication.

Reviewer 3 Report

This review makes a comprehensive summary of ALD and MLD coatings applied to alkali metal batteries in recent years. The process recipes of ALD and MLD, as well as the roles of coatings on enhanced electrochemical performance for Li, Na and K metals are reviewed. The authors have also presented the challenges for further researches of ALD and MLD on alkali metals, such as new precursors, novel ALD/MLD processes and in situ characterization methods. This review can be accepted after the following comments being considered.

1. Are the substrates in the Table 1 and 2 the same for Li or Na metal? If not, please list the substrates in the tables, which may be related to the process recipes.

 2. Section 3.1.2 and 3.1.3 state that fluoride and lithium phosphate coatings play very significant roles. Some performance curves could be presented for the readership.

 3. In Section 4, MLD is introduced in a simple manner, which has not been widely studied. It could be better if a comparison between MLD and ALD technology should be made here, and some explains on the advantages it has compared with ALD.

 4. The mechanism of ALD and MLD coatings for the enhanced electrochemical performance shall be discussed in the conclusion section.

 5. The author should give a perspective on the practical applications of some promising ALD and MLD coatings and discuss the challenges need to be overcome.
